# Evidence of High-Intensity Exercise on Lower Limb Functional Outcomes and Safety in Acute and Subacute Stroke Population: A Systematic Review

**DOI:** 10.3390/ijerph20010153

**Published:** 2022-12-22

**Authors:** Shi Min Mah, Alicia M. Goodwill, Hui Chueng Seow, Wei-Peng Teo

**Affiliations:** 1Physical Education and Sports Science Academic Group, National Institute of Education, Nanyang Technological University, Singapore 639798, Singapore; 2Department of Physiotherapy, Sengkang General Hospital, Singapore 544886, Singapore; 3Department of Physiotherapy, Singapore General Hospital, Singapore 168753, Singapore

**Keywords:** high-intensity, vigorous exercise, acute stroke, in-patient, rehabilitation

## Abstract

This systematic review investigated the effects of high-intensity exercise (HIE) on lower limb (LL) function in acute and subacute stroke patients. A systematic electronic search was performed in PubMed, CINAHL and the Web of Science from inception to 30 June 2022. Outcomes examined included LL function and measures of activities of daily living such as the Barthel index, 6 min walk test (6MWT), gait speed and Berg balance scale (BBS), adverse events and safety outcomes. The methodological quality and the quality of evidence for each study was assessed using the PEDro scale and the Risk of Bias 2 tool (RoB 2). HIE was defined as achieving at least 60% of the heart rate reserve (HRR) or VO_2_ peak, 70% of maximal heart rate (HR_max_), or attaining a score of 14 or more on the rate of perceived exertion Borg scale (6–20 rating scale). This study included randomized controlled trials (RCTs) which compared an intervention group of HIE to a control group of lower intensity exercise, or no intervention. All participants were in the acute (0–3 months) and subacute (3–6 months) stages of stroke recovery. Studies were excluded if they were not RCTs, included participants from a different stage of stroke recovery, or if the intervention did not meet the pre-defined HIE criteria. Overall, seven studies were included that used either high-intensity treadmill walking, stepping, cycling or overground walking exercises compared to either a low-intensity exercise (*n* = 4) or passive control condition (*n* = 3). Three studies reported significant improvements in 6MWT and gait speed performance, while only one showed improved BBS scores. No major adverse events were reported, although minor incidents were reported in only one study. This systematic review showed that HIE improved LL functional task performance, namely the 6MWT and gait speed. Previously, there was limited research demonstrating the efficacy of HIE early after stroke. This systematic review provides evidence that HIE may improve LL function with no significant adverse events report for stroke patients in their acute and subacute rehabilitation stages. Hence, HIE should be considered for implementation in this population, taking into account the possible benefits in terms of functional outcomes, as compared to lower intensity interventions.

## 1. Introduction

Stroke is one of the leading causes of adult disability locally and globally, and is Singapore’s fourth leading cause of death, accounting for close to 6% of deaths [1]. The prevalence of stroke is 3.65% among adults aged 50 and above, and the burden of post-stroke care is expected to rise due to the rapidly ageing population in Singapore [2,3]. Physical activity has been proven to be beneficial in the primary prevention of stroke; moreover, participation in physical activity may result in a reduction in the risk of recurrent stroke [4] and augment functional recovery [5], which may translate greater abilities to perform activities of daily living and less reliance on the healthcare system. As compared with older adults without stroke, people with stroke average 79% fewer steps per day [6,7] (1536–3035 vs. 7250 steps/day), far below that in the ‘‘sedentary lifestyle index’’ (5000 steps/day) [6]. Consequently, the resultant physical inactivity contributes to further de-conditioning [8], recurrent stroke [9], and high long-term risk for cardiovascular diseases [10]. Hence, targeted physical activity and exercise is necessary to drive functional recovery after stroke [11] which will lead to improved performance in functional activities.

The American Heart Association and American Stroke Association currently recommend aerobic exercise training for stroke survivors at least 3 days a week for 20–60 min per session, subject to the individual’s functional capacity, and resistance training 2 to 3 days a week, in the subacute and chronic stages of stroke recovery [12,13]. However, studies have shown that time spent performing physical activity and engagement in routine physiotherapy in individuals with sub-acute and chronic stroke is far below these optimal levels [11]. Stroke survivors are particularly predisposed to physical inactivity and a sedentary lifestyle, even in the acute stages within their inpatient stay because of residual physical disabilities, such as decreased mobility, poor balance, and muscle weakness [13]. A vicious cycle of physical inactivity is thus established where this sedentary behavior potentially affects motor recovery, perpetuates further deconditioning and contributes to long-term risk of recurrent stroke and cardiac events [14].

Exercise exerts multiple benefits on the cardiovascular, cardiorespiratory, and neurovascular systems in both healthy and clinical populations. Notably, exercise has been shown to augment neuroplasticity, resulting in functional improvements to motor [15] and cognitive abilities [16,17]. Thus, it is reasonable to suggest that exercise can be used as a viable neurorehabilitation strategy to remediate behavioral or functional outcomes associated with neurodegenerative disorders [18] or stroke [19]. In a recently published review by Moore [20] looking at current evidence in walking recovery after stroke, it was reinforced that there is limited research that is currently available to guide clinicians who are deciding between moderate- and high-intensity exercise for stroke survivors. It was mentioned that preliminary evidence that HIE could lead to better walking outcomes as compared to moderate intensity, but a full evaluation of the risk–benefit ratio of HIE is necessary and other means of attaining high-intensity post-stroke exercise, such as locomotor high-intensity interval training, should be further explored.

There is evidence that high-intensity exercise (HIE) can promote neuroplasticity and elicit significant improvements in aerobic fitness and peak capacity, as well as lower limb (LL) motor function (e.g., gait efficiency, walking speed and endurance) in persons post-stroke, when compared to lower intensity exercise [21]. In addition, a study by Holleran, Rodriguez [22] demonstrated that stepping training at high intensities was well-tolerated by stroke survivors in the subacute and chronic stages of their recovery, and resulted in significant locomotor improvements, as marked by increased gait speed, improved paretic single-limb stance, and increased walking endurance. However, exercising stroke patients at high intensities has given rise to safety concerns given that vigorous exercise can be associated with increased risk of cardiovascular events [23], especially in the early stages after stroke. Several systematic reviews and meta-analyses investigating the effects of HIE in stroke have suggested that the incidence of adverse events or acute cardiovascular events is low, with known improvements in outcomes in function, cardiovascular health, and neuroplastic measures [24,25]. However, these reviews have focused predominately on subacute and chronic stages post-stroke, with limited evidence available for the efficacy and safety of HIE in the acute stages of stroke.

From a neuroplastic standpoint, a cascade of events at the molecular, biological, and systemic level occurs almost immediately after stroke to limit any further damage to brain and initiate mechanisms for recovery [26]. These neuroplastic events help to spur spontaneous recovery of motor function, which some have purported to be a “golden window” of opportunity for neurorehabilitation in hopes of leveraging on this heightened phase of neuroplasticity [27]. However, the evidence for early interventions remains inconclusive. For example, results from the A Very Early Rehabilitation Trial (AVERT) in over 2000 stroke patients have reported that very early mobilization (within 24 h) led to reduced odds of favorable outcomes compared to the usual care control group [28], and these results cautioned clinicians against very early high-dose mobilization post-stroke. In another study, where early intervention was introduced at around 48 h post-stroke, clinical motor assessment (e.g., the Fugl–Meyer Upper Limb Assessment [FM-UL] and Barthel Index [BI]) outcomes were significantly improved [29]. These contradicting findings were indicative of how optimal dosage and intensity for improved outcomes in stroke were unclear, particularly in the early stages of recovery.

Therefore, to shed light on the role of high-intensity early neurorehabilitation in stroke, this systematic review aims to answer the following research questions:

(1) Is HIE effective for improving LL functional outcomes in the acute and subacute stroke population (0–6 months post-stroke)?

(2) Is the implementation of HIE safe in acute and subacute stroke settings?

The findings from this systematic review will provide support for the role of HIE in early stroke rehabilitation and provide clinicians with safety guidelines for its implementation.

## 2. Methods

This systematic review and analysis were carried out and presented according to the revised Preferred Reporting Items for Systematic Reviews and Meta-Analyses (PRISMA) guidelines [30], and the review protocol was registered in PROSPERO (CRD42022292683). Details of the definitions, search strategy, inclusion and exclusion criteria, risk of bias and quality of evidence, primary and secondary outcomes, data extraction, and statistical analysis are included below.

### 2.1. Search Strategy

A systematic search for articles in the databases PubMed, Web of Science and CINAHL (EBSCOhost) was conducted up until 30 June 2022. The keywords used for the search and strategies used may be found in Appendix A.

We included articles that fulfilled the following criteria.

(1)Randomized controlled trials (RCTs) which fulfilled the following criteria:
Active or passive control group as comparator;Adults aged 18 years and above;Any form of stroke;Stage of stroke: Acute (0–3 months) and subacute (3–6 months).
(2)Articles had to include a detailed description of the exercise intervention;(3)The intervention group had to meet the target of HIE;(4)The control group had to have a lower intensity training, or no exercise intervention;(5)All studies had to include a LL functional outcome measure.

Our exclusion criteria included unpublished and ongoing trials, and articles that were not published in English. The following sections describe the details of our inclusion criteria.

### 2.2. Types of Studies

This review included randomized controlled trials that included an intervention group receiving HIE and a control group receiving a physical therapy intervention of lower intensity or no specific intervention/usual care. Trials which involved more than one intervention group and crossover trials (data for the initial treatment were considered for inclusion due to the potential for long-term treatment effects to confound results) were included in this systematic review.

In addition, this review considered trials that evaluated the effectiveness of HIE in patients’ post-stroke. Vigorous or high-intensity was defined as achieving at least 60% of heart rate reserve (HRR) or VO_2_ peak, 70% of maximal heart rate (HR_max_), or attaining a score of 14 or more on the rate of the perceived exertion Borg scale (6–20 rating scale) in accordance with the American College of Sports Medicine’s guidelines for exercise prescription [3]. Exercise interventions included walking programs, task-oriented practice, or any other forms of training, performed at high intensity. Control interventions included no treatment, usual care, or low-intensity exercise regimes.

### 2.3. Types of Participants

This review considered studies investigating individuals over 18 years of age with a clinical diagnosis of acute stroke (as defined by American Heart Association/American Stroke Association as “an episode of acute neurological dysfunction presumed to be caused by ischemia or hemorrhage, persisting ≥24 h or until death”) [31] regardless of co-morbidities, duration since onset, severity of impairments, presence of preceding strokes or stroke location. The Stroke Recovery and Rehabilitation Roundtable taskforce defined the four stages following the onset as follows:(1)Hyper-acute or acute phase (0–7 days);(2)Early sub-acute phase (7 days until 3 months);(3)Late sub-acute phase (3–6 months);(4)Chronic phase (>6 months).

All participants included in this study suffered their stroke within 6 months of recruitment into the respective studies.

### 2.4. Selection of Studies

Titles and abstracts of the references retrieved from the search to identify studies that satisfy the inclusion criteria were screened by two researchers (SMM and HCS) independently. Studies that did not fulfill the inclusion criteria were excluded and full text articles of potentially relevant content were retrieved. Any conflicts in study inclusion/exclusion during the title, abstract and full-text article screening were first resolved by discussion between the two researchers, and if no resolution was achieved, the recommendation of a third researcher (WPT) would be solicited. Reasons for exclusion of studies and the process of selection were recorded, and a Preferred Reporting Items for Systematic Reviews and Meta-Analyses (PRISMA) flow diagram has been presented (Figure 1).

### 2.5. Primary and Secondary Outcomes

One researcher (SMM) was involved in extracting the data from the included studies. The primary outcome measures for this review included LL functional motor outcome measures that reflect the extent of impairments post-stroke. These outcome measures included the 6 min walk test (6MWT), gait speed, steps per day, Berg Balance scale (BBS) and the Barthel index (BI). Secondary outcomes in this study included reports of adverse events that occurred because of the intervention (i.e., dropouts, injuries, pain, swelling and death).

### 2.6. Risk of Bias and Quality of Evidence

The methodological quality of each study was independently assessed by two authors, SM and HC, by using the Physiotherapy Evidence Database (PEDro) Scale (range 0–10). Studies with a PEDro score ≥6 were considered to be of high quality; 4 or 5, of moderate quality; and <4, of low quality (www.pedro.org.au, accessed on 27 June 2022).

The quality of evidence was assessed by using version 2 of the Cochrane risk-of-bias tool for randomized trials (RoB 2) to assess the risk of bias in the included studies in this systematic review [32]. RoB 2 is defined by a fixed set of domains of bias, focusing on different aspects of trial design, conduct, and reporting. A tool guide was utilized for the authors to make a judgement on each domain being at a “low” or “high” risk of bias or having “some concerns”. Within the guide itself, a series of signaling questions for each domain aim to gather information about features of the trial that are relevant to risk of bias. A proposed judgement about the risk of bias arising from each domain is generated by an algorithm, based on answers to the signaling questions. Based on the judgement criteria, review authors can downgrade a randomized trial to low or some concerns.

## 3. Results

### 3.1. Search Results

A flow chart detailing the study selection process is in Figure 1. Seven studies (944 participants) were eligible for inclusion in this systematic review. The initial searches returned a total of 739 non-duplicate articles. Titles and abstracts of 711 articles were assessed for suitability and did not meet the inclusion criteria of the systematic review, leading to the retrieval of 28 full-text articles. Of these, 21 did not fulfil the inclusion criteria and the remaining seven studies were analyzed.

### 3.2. Characteristics of Included Studies

The seven included studies were all RCTs published from 2010 onwards [33,34,35,36,37,38,39]. The participants were enrolled in the studies between 1 and 139.7 days since the onset of their stroke. Study inclusion criteria included participants with an acute or subacute stroke (refer to definition of duration in Section. Types of Participants), stable cardiovascular and cardiopulmonary status, no severe musculoskeletal problems or pain, no significant cognitive or communication issues that will interfere with rehabilitation. Of the seven included studies, two studies included patients in the acute to early subacute stage [36,37], two studies included patients in the early subacute stage [35,38], two studies included patients in the early to late subacute stage [34,39], and one study included patients in the late subacute stage [33]. The characteristics of the included studies have been summarized in Table 1.

### 3.3. Interventions

The details of the interventions from the included studies are presented in Table 1. All seven studies included functional training or movements, and high-intensity walking or stepping within the exercise regime for the intervention group. Two studies used treadmill training as a modality [34,39], while three studies utilized cycling, either stationary or outdoor cycling for their HIE intervention [35,36,37].

Four out of the seven studies [34,35,38,39] had control groups where they exercised at lower intensities. They did conventional physiotherapy, which consisted of functional training, balance, strengthening, stretching, seated group and walking exercises. The other three studies [33,36,37] used a passive control group that provided education and advice on post-stroke symptoms, medication, lifestyle changes, physical activity, and exercise.

In all studies, either the heart rate reserve (HRR), peak oxygen consumption (VO_2peak_), peak HR and/or rating-of-perceived exertion (RPE) Borg scale were used to define the intensity of the exercise. Three of the seven studies measured exercise intensity using the HRR or RPE Borg scale [33,34,39], while the other studies [35,36,37,38] used RPE or HR_max_, which had to be attained during the exercise session in the intervention groups.

The duration of each intervention sessions ranged from 11 to 60 min. One study by Tollár, Nagy [38] had a session twice a day for the intervention group. The frequency ranged from 2 to 6 times per week, most occurring 4 to 5 times per week (*n* = 4) [34,37,38,39]. The intervention length ranged from 4 to 12 weeks, with the intervention for four studies being 4 to 5 weeks [33,35,38,39], the rest being 10–12 weeks [34,36,37]. Interventions were either undertaken in outpatient centers (*n* = 3) [33,38,39], inpatient rehabilitation units (*n* = 2) [35,36], both (*n* = 1) [34] and one study was home-based [37].

### 3.4. Outcome Measures

The primary outcome measures and results of the studies are included in Table 1. The LL functional motor outcomes were extracted from the included studies. A total of four studies investigated 6MWT [34,35,36,38], three looked at BBS [33,34,38], three at maximal gait speed [34,36,39] and one at self-selected gait speed [34], one used physical activity as measured by total steps/day [37], and two studies investigated BI [33,38].

### 3.5. 6 min Walk Test (6MWT)

Four studies investigated the effects of HIE on 6MWT [34,35,36,38]. Three out of the four studies showed significant improvements in walking distance in the HIE group compared to the control [34,36,38], two of which had active control groups exercising at lower intensity [34,38].

### 3.6. Gait Speed

Three studies measured gait speed [34,36,39], one of which measured both maximal and self-selected gait speed [34], while the other two investigated either one of them. All RCTs found significant improvements in gait speed in the group performing HIE compared to the control groups, two of which had active controls doing lower intensity exercise [34,39]. Two out of the three studies [34,36] performed a longer-term follow-up and found that the improvements in the HIE group were retained at 3–6 months post-stroke compared to the controls.

### 3.7. Berg Balance Scale (BBS)

Only the study by Tollár, Nagy [38] found significant changes in BBS in the HIE group in comparison with the active control group. The other two studies [33,34] did not find any meaningful differences between the HIE and passive control [33], as well as between the HIE and active control group [34].

### 3.8. Barthel Index (BI)

Only two studies [33,38] looked at BI as one of the outcome measures. Tollár and Colleagues [38] noted improvements in the high-intensity group in comparison with an active control group post-intervention, whereas Holmgren, Lindström [33] found a significant difference between the intervention and passive comparator group only at 6 months follow-up.

### 3.9. Physical Activity (Total steps/day)

Only one study [37] measured physical activity (total steps/day) as the primary outcome measure. However, no significant difference was observed between the intervention and control groups.

### 3.10. Adverse Events

Overall, five RCTs [35,36,37,38,39] reported no adverse events (i.e., orthopedic injury, cardiovascular events) during the study period, no serious adverse events were reported in any participants during or after the training sessions. Only one study [34] detailed minor adverse events, including joint and back pain, and skin breakdown (e.g., bruises and scrapes). The study by Holmgren, Lindström [33] did not report any adverse events during the intervention but they recorded falls outside of the training sessions during the study period and found no significant differences regarding falls related injuries in both groups.

All studies reported good compliance and adherence to completion of the interventions and follow-up assessments. All studies had more than 90% of subjects completed the intervention program except in the study by Pallesen and Colleagues [35] where three subjects in the intervention, and five in the control group were not able to complete the prescribed training program due to medical reasons; the remaining 30 participants completed the 4 week program and the 3 month follow-up assessment. Attendance rate was reported to be between 79–100% in all studies, and the dropout rate for follow-up assessments was between 3.6–15.6%.

### 3.11. Quality of Included Studies

The methodological quality of each RCT is in Figure 2. All studies had a PEDro score of 5 and above, and hence are of moderate-to-high quality. All trials were randomized, and allocation was concealed in 6 out of the 7 studies [33,34,36,37,38,39]. None of the included studies had blinded therapists, or blinded participants, which is understandably difficult to achieve in physical therapy trials. However, all trials had blinded assessors who measured at least one of the key outcomes. Two out of the seven studies used an intent-to-treat analysis [33,34]. All studies presented the results of between-group characteristics and statistical comparisons, and baseline outcomes in the comparison groups were similar. In addition, all included studies reported both point measures and measures of variability for at least one key outcome and noted that more than 85% of the participants initially assigned to groups had at least one of the key outcomes obtained. Five studies were rated as high quality [33,34,36,37,38] and two studies were of moderate quality [35,39].

### 3.12. Risk of Bias

Based on the RoB 2 tool, the outcomes are presented in the Figure 2 below. Overall, the risk of bias assessment showed that four out of the seven included studies [35,37,38,39] had some concerns in terms of the defined domains of risk of bias in the aspects of trial design, conduct, and reporting.

## 4. Discussion

This systematic review investigated the effects of HIE in the acute and subacute stages of stroke, as defined by acute and subacute stages (up until 6 months) of rehabilitation post-stroke, on measures of LL functional motor outcomes, safety, and adherence. In total, seven studies with 944 participants met our inclusion criteria for this systematic review. Our results indicate that five out of the seven RCTs reported improved LL functional outcomes in the HIE group, as compared to the control group, of which three had active comparator groups exercising at lower intensity [34,38,39], and the remaining two studies had passive control groups which received only education and general advice [33,36]. Different types of HIE interventions were presented in this systematic review, and due to the limited number of studies and varied types of HIE interventions, the effectiveness of these specific interventions cannot be fully established, particularly in the early stages of recovery after stroke. The purpose of this review was to look at the efficacy and safety of HIE in the first 6 months post-stroke, so that the results can be translated into actionable initiatives in a clinical setting. In addition, the varied outcome measures used in the different studies resulted in insufficient quantitative data to perform a meta-analysis. Nevertheless, only one study [34] reported minor adverse events, and all studies reported high compliance and good adherence (>90% completion) of the interventions and follow-up assessments. The attendance rate for the HIE interventions and the control groups was reported to be 79% and above in all studies, and the dropout rate for follow-up assessments was low. Overall, our findings suggest that HIE tailored to individuals with stroke is feasible and safe and may result in improvements in functional abilities in acute stroke patients.

### 4.1. Effects of HIE on Stroke Recovery

This systematic review, supported by similar reviews that compared HIE to control groups in other stages of stroke recovery [25], has shown that HIE is likely to be beneficial in improving LL function in acute and subacute stroke patients. The improvement in LL motor outcomes in the HIE intervention groups is likely to be underpinned by neuroplastic mechanisms induced by exercise that includes an increase in cerebral blood flow, upregulation of neurotrophin expression (i.e., brain-derived neurotrophic factor [BDNF], Insulin-like growth factor 1 [IGF-1] and vascular endothelial growth factor [VEGF]) and improvements in cardiometabolic function (i.e., greater blood sugar regulation and fat metabolism) [16,19,40]. However, most studies thus far have often focused on stroke patients in the chronic stroke phase (>6 months post-stroke) or have used exercise intensities that are within the mild-to-moderate (working up to and not more than 59% HRR or Peak VO_2_ or 69% Peak HR or 13 on RPE Borg Scale) range [41].

Due to the physical impairments after stroke, stroke survivors may have difficulty reaching and sustaining high-intensity in their exercise regimes [21,42], hence, the type of HIE intervention prescribed for this population is important. The literature has begun to show that the modalities interventions, such as high-intensity interval training (HIIT) may be better tolerated in this population [20]. This may help to overcome the barriers of stroke survivors, such as central and local fatigue, which is at present not well understood [43], and enable participants to reach the required exercise intensity for optimal functional gains.

Indeed, our findings showed that five out of seven studies included in this systematic review reported significant improvements in LL functional outcomes following a period of HIE intervention compared to the control group, in acute to subacute stroke patients. The improvements in functional outcome included improved walking distance, gait speed, and BI (measure of function during activities of daily living), with gait speed and BI showing sustained improvements up to 6 months post-intervention. It should however be noted that the included studies focused predominantly on lower limb exercises such as high-intensity stepping, walking, cycling that maybe complimented with functional training, which is likely to explain why a larger proportion of studies reported improvements in 6MWT (three out of four studies) and gait speed (three out of three studies) as compared to BBS (only one out of three studies).

Our findings were in-line with a previous meta-analysis (of 22 included studies) conducted by Luo, Zhu [25] showing that HIE was effective in improving measures of walking such as walking distance, stride length, comfortable walking speed and timed-up-and-go in patients with subacute and chronic stroke. Similarly, Ward [44] demonstrated in a large cohort of chronic stroke patients (*n* = 224, 18 months median time post-stroke) that high-intensity Upper Limb rehabilitation improved clinical measures of Upper Limb motor function (i.e., Fugl–Meyer Upper Limb [FM-UL], Action Research Arm Test [ARAT], and the Chedoke Arm and Hand Activity Inventory), with 68.3% and 61.6% of patients reaching minimal clinical important difference in the FM-UL and ARAT, respectively even at 6 months post-intervention. Other studies utilizing various high-intensity training modalities such as resistance training [45], robotic training [46] and stepping training [22] have showed similar improvements in upper and lower limb motor outcomes, providing evidential support for the role of HIE in improving motor outcomes for stroke patients in the acute to chronic phase.

### 4.2. Safety of HIE in Stroke Rehabilitation

The use of HIE, particularly in populations with a higher risk of cardiovascular events, has often been limited, in part, due to the fear of potentially causing further adverse events from occurring. Additionally, patients may have difficulties in reaching high levels of exercise intensities, particularly when movement may be affected [20]. Looking at a different population, a meta-analysis in adults with coronary artery disease or heart failure showed that the risk of any cardiovascular event is very low and appears to be safe [47]. Regarding the safety of HIE in the acute and subacute stroke population, our systematic review showed that all seven studies reported good adherence to the protocol with no reports of dropouts. Only one study reported minor adverse events (i.e., muscle aches and pains) [34].

Even though minimal safety concerns or adverse events were reported across all studies, precautions still need to be taken to minimize risks to patients. Marzolini, Robertson [23] proposed several strategies to protect the brain from potential adverse circulatory effects before, during and after mobilization and exercise sessions in people with stroke. Given that there is a possibility of serious cardiac adverse events in the acute post-stroke phase, clinicians need to be aware of the complex nature of stroke presentations and the neurobiological and physiological changes that occur with exercise. Pre-participation and cardiac screening criteria are also necessary prior to enrolment into the exercise programmes.

### 4.3. Limitations

There are several limitations of this systematic review that we would like to acknowledge. Firstly, four out of the seven studies performed follow-up 2-6 months post-intervention, and this is insufficient to fully justify the improvements and effectiveness of the HIE early after stroke. Additionally, the inclusion of only seven papers that fit into our inclusion criteria is insufficient to draw any conclusions on dose-/intensity-response relationships on motor recovery following stroke. To better understand and provide evidence-based interventions of sufficient intensity to ensure optimal outcomes post-stroke, further studies need to recruit a bigger clinical population, standardize outcome measurements across studies investigating different intensities, and ensure adequate long-term follow-up. Secondly, the variations in training parameters, specifically the type of HIE rendered which varied from stepping exercises to ergometer cycling to gaming, and the types of outcome measures included in the studies were highly variable. This makes it difficult to make recommendations on the type of HIE that stroke survivors can perform and be certain if HIE will result in benefits in specific functional outcomes. Most studies to date of HIE in stroke have used either treadmill or overground stepping and walking as the mode of training, but research is emerging on other modes of HIT, including seated modalities such as the recumbent stepper and high-intensity interval training. This offers opportunities for individuals with a range of functional abilities to be included in studies investigating HIE.

## 5. Conclusions

Our results show that HIE in acute and subacute stroke patients may be effective in improving motor function of the lower and upper limb. In addition, the included studies demonstrate that supervised HIE can be safely administered with good adherence and minor adverse events, such as muscle aches and pains. However, due to the limited studies included, the findings from these studies shown need to be interpreted with caution. Further, most of the literature available do not differentiate between the stages of stroke, and studies include all patients independent of the time since onset of stroke. As a result, very little evidence is available looking at the efficacy of higher intensity exercise early after stroke, and this then does not allow us to make proper recommendations when prescribing exercise for this population. More work in this area needs to be done, so that we can fully optimise neuro-recovery in the different stages of stroke recovery throughout their rehabilitation.

## Figures and Tables

**Figure 1 ijerph-20-00153-f001:**
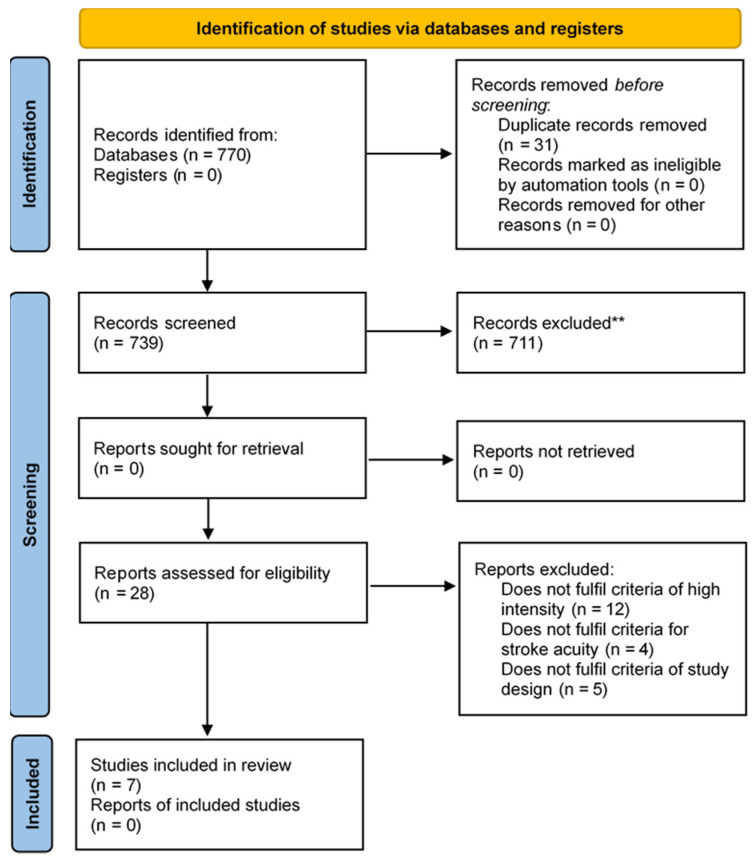
PRISMA flow diagram of study selection process. A total of seven RCTs fulfilled the selection criteria and were included in this review (adapted from Page et al. [30]).

**Figure 2 ijerph-20-00153-f002:**
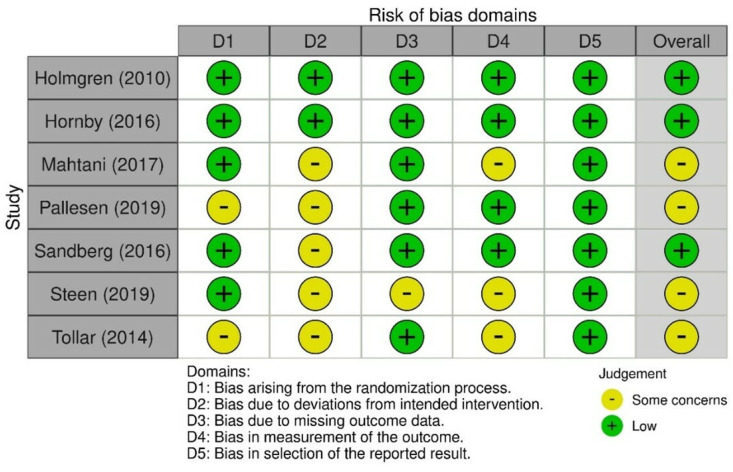
A consolidation of ROB2 outcomes in the included studies, adapted from: https://www.riskofbias.info/welcome/rob-2-0-tool/current-version-of-rob-2 (accessed on 20 October 2022).

**Table 1 ijerph-20-00153-t001:** Summary information for all included studies in this systematic review.

**Holmgren (2010)** Number of stroke participants: IG—15 CG—19 Age of study participants, Mean (±SD, Years): IG—77.7 ± 7.6 CG—79.2 ± 7.5 Duration since stroke, Mean (±SD): IG—139.7 ± 37.3 days CG—126.8 ± 28.2 days Inclusion criteria: First ever or recurrent ischemic or hemorrhagic stroke 3–6 months before enrollment and randomization Age ≥ 55 Ability to walk 10 m with or without a walking device Ability to understand and comply with instructions in Swedish Subjects should have risk of fall at the time of enrolment Exclusion criteria: Ability to walk outdoors independently i.e., without personal assistance or walking device Severe aphasia or severe vision or hearing impairment A medical condition that a physician determined was inconsistent with study participation, e.g., cancer or severe congestive heart failure with expected short remaining life expectancy A recurrent stroke within 3 months before study starts If the individual lived more than 100 km away from the training facilities, this was considered too far away IG: Frequency—30 sessions over 5 weeks Intensity—70–80% HRR or RPE 15–17 Type—High-intensity functional exercises targeting strength, function and gait Time—1.5 h Setting—Outpatient clinic CG: Frequency—1 session weekly for 5 weeks Intensity—NA Type—Educational sessions on post-stroke symptoms Time—1 h Setting—Outpatient clinic PEDro score—8/10 Assessment timepoints and functional outcomes: Baseline, post-intervention, 3 months and 6 months follow-up Berg Balance Scale (BBS) Barthel Index (BI) Frenchay Activities Index (FAI-3) Results: There were no significant differences between the IG and the CG regarding BBS and FAI-3. At 6 months, significant difference in favor of IG for BI *p* = 0.05. At 6 months, significant difference in favor of IG for BI *p* = 0.05.
**Hornby (2016)** Number of stroke participants: IG—16 CG—17 Age of study participants, Mean (±SD, Years): IG—57 ± 12 CG—60 ± 9.2 Duration since stroke, Mean (±SD): IG—114 ± 56 days CG—89 ± 44 days Inclusion criteria: Individuals with a single, unilateral, supratentorial stroke in the previous 1 to 6 months All participants were required to walk 10 m overground with minimal or moderate physical assistance from a therapist or without physical assistance but at speeds ≤0.9 m/s at self-selected speeds (SSSs) with assistive devices and below, knee bracing as needed Exclusion criteria: Previous central or peripheral nervous system or orthopedic injury that may limit independent ambulation Uncontrolled cardiorespiratory disease Inability to follow three-step commands IG: Frequency—40 sessions over 10 weeks Intensity—70–80% HRR or RPE ≥14 Type—Treadmill and overground training, stairs climbing Time—1 h Setting—Inpatient and outpatient clinics CG: Frequency—40 sessions over 10 weeks Intensity—30–40% HRR Type—Conventional physiotherapy (functional training) Time—1 h Setting—Inpatient and outpatient clinics PEDro score—7/10 Assessment timepoints and functional outcomes: Baseline, mid-training, post-intervention and 2 months follow-up Gait speed Self-selected speed (SSS) Fastest speed (FS) 6 min walk test (6MWT) Berg balance scale (BBS) Steps/day Five times sit-to-stand (5XSTS) Physical SF-36 Results: Focused variable stepping training at high intensities elicits significantly greater gains in walking function and participation as compared with conventional interventions. The magnitudes of the differences in SSS and 6MWT between groups were above clinically significant thresholds. Standing balance and STS performance demonstrated equivalent gains in both groups. No significant differences in BBS, Steps/day and 5XSTS between groups Significant difference for SSS in favor of IG *p* = 0.002 Significant difference for FS in favor of IG *p* = 0.006 Significant difference for 6MWT in favor of IG *p* = 0.001. Significant difference for Physical SF-36 in favor of IG *p* = 0.014
**Mahtani (2017)** Number of stroke participants: IG—16 CG—17 Age of study participants, Mean (±SD, Years): IG—54 ± 12 CG—61 ± 9.3 Duration since stroke, Mean (±SD): IG—106 ± 57 days CG—88 ± 41 days Inclusion criteria: 18 to 75 years of age Able to walk 10 m overground with at least minimal or moderate Physical assistance from a therapist or without physical assistance, but at speeds of ≥0.9 m/s at self-selected walking speed. Participants were allowed to use assistive devices and below-knee bracing as needed Ability to walk at least 0.1 m/s on a motorized treadmill without weight support but with use of a handrail at baseline and post training Exclusion criteria: Previous central or peripheral nervous system or orthopedic injury that may limit independent ambulation Inability to ambulate independently at least 45.7 m prior to stroke Uncontrolled cardiorespiratory disease Inability to follow three-step commands or adhere to study requirements IG: Frequency—40 sessions over 10 weeks Intensity—70–80% HRR or RPE 15–17 Type—Treadmill training, and continuous stepping practice in multiple, variable environments Time—1 h Setting—Inpatient and outpatient clinics CG: Frequency—40 sessions over 10 weeks Intensity—30–40% HRR Type—Conventional PT (functional training) Time—1 h Setting—Inpatient and outpatient clinics PEDro score—5/10 Assessment timepoints and functional outcomes: Baseline and post-intervention Peak gait speed Results: Significant improvements in peak speed favoring IG over CG, 0.48 ± 0.31 versus 0.13 ± 0.17 m/s respectively, *p* < 0.01.
**Pallesen (2019)** Number of stroke participants: IG—16 CG—14 Age of study participants: IG—Median: 55, IQR: 50–60, Range: 43–67 CG—Median: 50, IQR: 44–56, Range: 22–64 Duration since stroke, Mean (±SD): Only stated that participants were recruited 1–3 months post-stroke Inclusion criteria: Sub-acute stroke (1–3 months after incident) Age 20–70 Score of maximum 5 on at least two items on the cognitive subscales of FIM A minimum score of 45 on Berg Balance Scale (BBS) Length of hospitalization expected to be >4 weeks Exclusion criteria: Medical conditions (e.g., heart or lung diseases) that would restrict participation Not able to provide consent Severe aphasia Dementia or cognitive impairments diagnosis prior to stroke Enrolled in other projects Language barriers due to having a native language other than Danish IG: Frequency—2x/week, 4 weeks Intensity—>70% max HR Type—Stepping, walking, cycling and sit-to-stands Time—45 min Setting—Inpatient rehabilitation ward CG: Frequency—2x/week, 4 weeks Intensity—<60% max HR Type—Conventional physiotherapy (functional training) Time—45 min Setting—Inpatient rehabilitation ward PEDro score—5/10 Assessment timepoints and functional outcomes: Baseline, post-intervention and 3 months follow-up Aerobic capacity and endurance: 6 min walk test (6MWT) Results: The group differences were not statistically significant for 6MWT *p* = 0.12. The IG improved on average by 73 m and the CG by 37 m. At 3-months follow-up, both groups had improved further; the IG another 21 m and the CG 33 m.
**Sandberg (2016)** Number of stroke participants: IG—29 CG—27 Age of study participants, Mean (±SD, Years): IG—71.3 ± 7.0 CG—70.4 ± 8.1 Duration since stroke, Mean (±SD): IG—22.2 ± 10.1 days CG—22.8 ± 10.8 days Inclusion criteria: All subjects had a stroke that was diagnosed by a physician within 3 days prior to the request for inclusion Subjects had to be able to walk > 5 m with or without support Able to understand spoken and written instructions Their impairments corresponded to mild stroke (National Institutes of Health Stroke Scale score < 6) Exclusion criteria: Medical or neurologic diseases that could either be a risk or make the training program difficult to fulfil. This judgment was made by the treating physician. IG: Frequency—2x/week, 12 weeks Intensity—80% max HR or RPE 14–15 Type—Ergometer cycle + functional training Time—1 h Setting—Inpatient stroke unit CG: Frequency—Nil active intervention Intensity—NA Type—General advice on physical training and activity and encouraged to return to previous activity levels Time—NA Setting—Inpatient stroke unit PEDro score—7/10 Assessment timepoints and functional outcomes: Baseline, post-intervention and 6 months follow-up 6 min walk test (6MWT) Maximal gait speed Timed up and go (TUG) Results: Intensive aerobic exercise twice weekly early in subacute mild stroke improved aerobic capacity, walking and balance. Significant difference for 6MWT in favour of IG *p* = 0.011 Significant difference for maximal gait speed in favor of IG *p* = 0.003 Significant difference for TUG in favor of IG *p* = 0.044
**Steen (2019)** Number of stroke participants: IG—31 CG—32 Age of study participants, Mean (±SD, Years): IG—63.7 ± 8.9 CG—63.7 ± 9.2 Duration since stroke, Mean (±SD): None specified, however patients were enrolled within 1–21 days of stroke onset. Inclusion criteria: Patients 18 years or older diagnosed with a first-time lacunar stroke or a recurrent event of lacunar stroke were enrolled in the study Patients with clinical symptoms with a verified relevant brain stem or subcortical hemispheric lesion (<2 cm in diameter in the acute phase) based on computed tomography (CT) scan or magnetic resonance imaging (MRI) scan Patients had a severity of neurological symptoms, categorized as “mild” on the Scandinavian Stroke Scale (SSS) (43–58 points) Patients had to speak and read Danish and provide informed consent Exclusion criteria: Patients with previous large-artery stroke, unstable cardiac condition, atrial fibrillation, pacemaker, uncontrolled hypertension, uncontrolled diabetes, artery stenosis >50%, symptoms or comorbidities not allowing exercise on a stationary bicycle Dyspnea caused by heart or pulmonary disease Aphasia, or dementia that interfered with understanding the protocol and physical examinations. IG: Frequency—5x/week, 12 weeks Intensity—77–93% max HR or RPE 14–16 Type—Stationary/outdoor cycling, stair climbing, brisk walking or running Time—HIIT 3 × 3 min, 2 min active recovery Setting—Home based CG: Frequency—Nil active intervention Intensity—NA Type—Advice on medication, prevention, lifestyle changes, resuming physical activity and physical activity tracking Time—NA Setting—Home-based PEDro score—7/10 Assessment timepoints and functional outcomes: Baseline and post-intervention Physical activity (total steps/day) Results: High Intensity Interval Training (HIIT) was feasible and safe in patients with lacunar stroke. Between-group differences in total steps/day, was not statistically significant *p* = 0.8.
**Tollar (2014)** Number of stroke participants: IG 1—286 IG 2—272 CG—83 Age of study participants, Mean (±SD, Years): IG 1—67.6 ± 5.49 years IG 2—65.9 ± 6.10 years CG—64.9 ± 5.80 years Duration since stroke, Mean (±SD): IG 1—2.8 ± 0.74 weeks IG 2—2.9 ± 0.77 weeks CG—2.8 ± 0.72 weeks Inclusion criteria: First-ever ischemic stroke diagnosed by a neurologist based on computed tomography or magnetic resonance imaging scans Time after stroke of 2 to 4 weeks Mobility and postural limitation determined by neurologic exam, and a modified Rankin Scale (mRS) score of 2 or higher Exclusion criteria: A history of multiple strokes Systolic resting blood pressure (rSBP) less than 120 or greater than 160 mmHg Orthostatic hypotension, carotid artery stenosis, severe heart disease, hemophilia, traumatic brain injury, seizure disorder, uncontrolled diabetes, abnormal electro-encephalography Mini Mental State Examination score less than 22 Abnormal blood panel Use of sedatives, irregular medication schedule Severe aphasia (Western Aphasia Battery, 25) Severe visual or hearing impairments, sensory dysfunction Severe orthopedic problems, neurologic conditions affecting motor function (PD, multiple sclerosis, multiple system atrophy, Guillaine Barre’ syndrome) Alcoholism, recreational drug use, smoking after stroke diagnosis Inability to walk a minimum of 100 m with or without a walking aid in 6 min Berg Balance Scale (BBS) score of 32 or less Barthel Index (BI) score of 70 or less Inability to understand verbal instructions or prompts from a television screen, or current participation in a self directed or formal group exercise program other than standard physical therapy IG: Frequency IG 1—5x/week, 5 weeks, IG 2—5x double sessions/week, 5 weeks Intensity—RPE 14–16 Type—Stepping, gait, agility training and Xbox gaming Time—1 h per session Setting—Outpatient gym CG: Frequency—5x/week, 5 weeks Intensity—Lower intensity, not defined Type—30 min seated group exercises and 30 min walking and balance exercises Time—1 h per session Setting—Outpatient PT gym PEDro score—7/10 Assessment timepoints and functional outcomes: Baseline and post-intervention Modified Rankin Scale (mRS) Barthel Index (BI) 6 min walk test (6MWT) Berg balance scale (BBS) Results: Twice daily HIE produced superior effects on clinical and motor symptoms, BP, and QoL in subacute ischemic stroke participants compared to the other groups. Effects of interventions on outcomes (Mean ±SD) 1. mRS IG 2: −1.8 ± 0.81 IG 1: −1.4 ± 0.95 Control: −0.7 ± 0.73 Improvements of 1.8 (IG 2) and 1.4 (IG 1) in mRS, the primary outcome, exceeded the clinically meaningful change of 1 unit 2. BI IG 2: 27.2 ± 8.92 IG 1: 19.2 ± 12.37 CG: 10.3 ± 7.94 The BI measures the perceived ability to perform activities of daily living, which improved by 8 to 9 points, strongly favoring exercise intensity and frequency versus standard care 3. 6MWT IG 2: 124.3 ± 78.13 m IG 1: 108.6 ± 74.32 m CG: 62.8 ± 66.25 m Participants in IG 2 groups walked approximately 125 m further than at baseline. This gain was nearly twice the gain in the lower frequency (IG 1) and intensity (CG) groups 4. BBS IG 2: 6.8 ± 6.28 IG 1: 4.6 ± 6.29 CG: 2.8 ± 4.51 Significant improvements were also seen in the intervention groups in BBS as compared to the control group.

Legend: IG: Intervention Group, CG: Control Group, SD: Standard Deviation, HRR: Heart Rate Reserve, RPE; Rate of Perceived Exertion, HR: Heart Rate, NA: Not Applicable, M: meters, m/s: meter/ second, IQR: Inter-Quartile Range.

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
