# Peer review of "Evidence of High-Intensity Exercise on Lower Limb Functional Outcomes and Safety in Acute and Subacute Stroke Population: A Systematic Review"

_ijerph, 2022, doi:10.3390/ijerph20010153_

Round 1

Reviewer 1 Report

This manuscript is a review article on 7 publications on the effect of high intensity exercise on lower limb function in the first 6 month after acute stroke. Overall, the authors found that although most publications reported an improvement in lower limb function, the studies were heterogenous regarding their endpoints for measuring lower limb function and also regarding the type of exercise used in the rehabilitation program. This makes it difficult to perform a direct comparison of the different studies.

The strength of this review is a comprehensive description of the study design and the results of the 7 most relevant studies on high intensity exercise. 

The weakness of this review is that because of different endpoints it was not possible to make a direct comparison of the studies. 

Recommendations:

The tables 2 and 3 are difficult to read because there is too many information presented in narrow columns. It would be more comfortable for the readers to have the information of the tables presented in the text body with one paragraph for each study.

For example:

Holmgren (2010)

Number of stroke participants: …

Age of study participants: …

Duration since stroke: …

Inclusion criteria: …

Exclusion criteria …

Intervention group: …

Control Group: …

PEDro score:

Hornby (2016)

and so on. 

Author Response

We have now amended the table into the stipulated format requested by the review (Please see Table 2).

Reviewer 2 Report

This study used a systematic review to gather evidence of the effectiveness of high intensity exercise on lower-limb functional outcomes in an acute and subacute stroke population. The paper is well written but several details need to be expanded upon in order to understand the methodological approach and implications of the findings. Line by line suggestions are given below.

Line 13: Should say “Tests for LL function and activities of daily living…”

Line 17: What was your definition for high-intensity training? What were the exact inclusion and exclusion criteria for paper retrieval? Please include these details here.

Line 19: Start a sentence with number written out: Three studies reported….

Line 21. What was the research question to be addressed by this review? The effect of HIE or the resultant adverse effects that are associated with HIE? The purpose of performing this review is not clear.

Line 21-22: Sentence is not clear. Please re-write. “… HIE across functional task performance namely in the 6-minute walk test...”

Line 23: Please include a rationale and definition for acute and subacute stroke. What was the purpose of limiting the search to these studies only?

Line 30: Which prevalence are you referring to? Prevalence of death or stroke in general? Please clarify

Line 45-47: Is this recommendation in the subacute population? Or for chronic stroke survivors?

Line 57-61: Statement here makes it sound like this is the only evidence of the effect of HIE on motor function however many studies have examined this. Please add to this statement.

Line 63-64: “… as marked by increased gait speed, ….., and increased walking endurance”

Line 80-81: Sentence structure needs clarifying. “… (AVERT) in over 2000 stroke patients has reported….”

Line 89: Please define acute and subacute stroke.

Line 91: add the word review

Line 92: Please clarify “neurorehabilitation stroke”

Line 130: should be Table 1 – not table 1.

Line 148: Any data on the average chronicity of participants from papers included in your study? Even within the 6-month acute window there is a lot of potential for change.

Line 160: What was the main reason for excluding papers and reducing the original 739 records by 711? Additional details are needed on reasons for exclusion and a justification for only including 7 studies in the review.

Line 166-168: Were these outcome measures chosen apriori before the review or were they chosen as the most prevalent measures from the included articles?

Line 213,253,258, 263, etc.: Please start the sentence with the number written out: Four of the 7 studies….

Line 238: Table 2 seems rather large. Information that is presented in the text may not need to be duplicated in the table and vice versa.

Line 241: Were any other LL functional motor outcomes measured in the included studies?

Line 249: Again, information is redundant from the text to Table 3 and redundant again from lines 251 to 277.

Line 251: Delete the period after the 6 – “6-minute walk test”

Line 332-334: Details are needed on why studies were excluded. As this was a main purpose of completing the systematic review, why weren’t more studies included in order to make inferences about the effectiveness of HIE interventions on outcomes. What are we gaining in this review?

Line 340: Add the word “in” - May results in improvements in functional …

Lines 342-356: This paragraph seems better suited for the Introduction as it’s not a discussion based on your results and is general background information on exercise in clinical populations.

Line 354: Please speculate on what the associated risks would be with conducting a risk-benefit analysis for HIE. From your data, it seems there are very few risks since there were very few adverse events and high participation rates.

Line 358: How is your review unique or different compared to the Luo et al. 2019 review? Acute stroke participants?

Line 400: Add the word “as” – training modalities such as resistance training …

Line 413: capital letter needed on the word “Only”.

Line 404-414: Given that there are few adverse events or safety concerns regarding HIE in the acute and sub-acute stroke population, what may be your recommendation regarding high intensity exercise in this clinical population?

Line 419: Why were more papers not included then?

Line 424-425: Perhaps more discussion is needed around this idea of variability in training interventions. Are there data on the cardiovascular implications of these training modalities to at least compare intensity between types of exercise? Even from the data presented here, how do RPE scores compare across exercise modalities? If they are similar then does the mode of exercise even matter?

Round 2

Reviewer 2 Report

All comments have been adequately responded to and addressed with this revision. The purpose for performing the review and interpretation of the outcomes of the review has been clarified and strengthened with these additional details and explanations. I have no further comments.

Author Response

Thanks.